# Beneficial Effect of Pre-Hardening of Elements Manufactured by the SLA Technology

Bartosz Pszczółkowski [1], Magdalena Lemecha [1], Wojciech Rejmer [1], Krzysztof Ligier [1], Mirosław Bramowicz [1], Magdalena Zaborowska [2] and Sławomir Kulesza [1,*]

[1] Faculty of Technical Sciences, University of Warmia and Mazury in Olsztyn, Oczapowskiego 11, 10-719 Olsztyn, Poland
[2] Faculty of Geoengineering, University of Warmia and Mazury in Olsztyn, 10-719 Olsztyn, Poland
* Correspondence: slawomir.kulesza@uwm.edu.pl

**Abstract:** In this work the effect of preliminary curing on mechanical, physicochemical and tribological properties of SLA (Stereolithography Appearance) manufactured samples is presented. Three preliminary curing times of 5, 10 and 15 s were selected for SLA manufacturing. The materials' friction, hardness and capacitance values were determined by ball cratering, Brinell method and electrochemical impedance spectroscopy, respectively. The obtained results showed that the mechanical property values changed most significantly after 10 s of preliminary curing, but friction wear and electrochemical capacitance showed greatest change for samples cured for 15 s. This effect may be explained by the domination of elongation of molecular chains in the first 10 seconds of preliminary curing and the gradual increase of branching processes during the next 15 s.

**Keywords:** stereolithography; polymers; hardness; wear; solidification

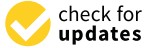



## 1. Introduction

Recently, rapid prototyping (RP) has become an indispensable method of manufacturing structures with high geometrical complexity, which are difficult, or even impossible, to fabricate using traditional manufacturing methods. Rapid prototyping techniques, such as fused deposition modeling (FDM) [1–3], selective laser sintering (SLS) [4], 3D printing (3DP) [2,5,6] and stereolithography appearance (SLA) [7], have already been used to produce polymeric and ceramic cellular solids for a large number of applications. Aside from these applications, various SLA modifications, such as the application of digital mask generators (e.g., liquid crystal displays and digital mirror devices, DMD), have also been successfully used for the construction of ceramic and polymeric structures [7–10].

Currently, in SLA technology, an element is manufactured through selective bonding of layers of photosensitive resin exposed to directed light beams. Incident light causes polymerization in the selected network. This process continues layer by layer, while the liquid resin fills a stationary vessel with low absorption foil at the bottom. During printing, the model is partially submerged in liquid resin. Submersion is limited by vessel size and amount of resin. After curing the single model layer, the platform rises, the model is disconnected from the bottom and the space between model and actuator is filled [11–14].

Recently, SLA has become increasingly popular as a low-budget additive manufacturing technology. It is well regarded by designers and engineers because of its high dimensional accuracy and higher element quality, in comparison to the more popular FDM (Fused Deposition Modeling) technology [15–17] Early instruments for 3D SLA printing were not as successful as FDM printers because of high costs of materials and low availability. Currently, most of the initial problems have been solved and producers offer a larger assortment of SLA resins for many various applications. Additionally, resins used in manufacturing are characterized by high light transparency, which allows their application in optics, and models can be further treated by mechanical means. The mechanical strength

of photocured models was demonstrated by Berger [18], who published research on the properties of gears obtained with SLA technology. It was observed that there was neither wear nor damage deterioration of printed bevel gears after work under 0.25 N·m torque and 120 rpm frequency. Having such high-performance parameters has opened up perspectives for their application in many industries, including medical technologies, where the elements are used to produce biocompatible implants and bone replacements [2].

## 2. Materials and Methods

In the presented research, disks of 25.4 mm in diameter and 10 mm thickness were used as samples. The elements were manufactured by means of SLA technology from the resin (brand name: Anycubic Standard Resin) exposed to the light in a range from 365 to 405 nm for 5, 10 and 15 s. Each layer of manufactured elements was 50 µm thick. The printing was carried out in air at room temperature (21 °C), but all samples were then, additionally, cured for 48 h at room temperature under sunlight conditions. The samples were produced using the "Anycubic®Photon-S" printer working in SLA technology based on LCD, (Liquid Crystal Display) with transparent resin obtained from the printer producer. The hardness of obtained materials was measured by means of "Innovatest Nexus 703A", using the Brinell method for the polymers. The penetrating force was 358 N and the dwelling time was 10 s. The hardness of each sample was measured at six different locations. Wear resistance of materials was measured using a T-20 abrasive testing machine [19], by means of the ball-cratering method according to PN-EN 1071-6:2007. The ball-cratering method is based on measurement of crater diameter. Craters are created during friction as a result of rotational movement of a ball in the presence of a friction agent. The scheme of the ball cratering method measurement is presented in Figure 1. Statistical significance of differences between hardness data upon curing time were made with the help of Statistica, version 13.3 (StatSoft).

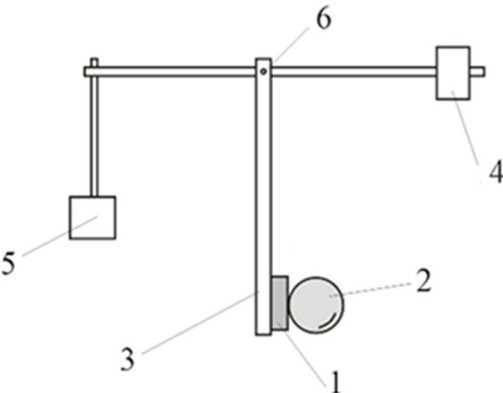

**Figure 1.** The scheme of ball cratering investigation: 1—sample, 2—test ball, 3—sample holder, 4—counterweight, 5—weight, 6—pivot.

The rotating ball (counter sample) was pressed with a given force to a sample surface, and friction agent was placed in between. The evaluation of the specific wear rate was estimated from the size of a crater. In our research, the ball 25.4 mm in diameter, with a surface roughness of Ra 0.177 µm was made from 100Cr6 steel with 58.6 HRC (hardness Rockwell scale C) hardness. The tests were performed without any friction agent in order to eliminate possible contamination of relatively soft samples with particles of the friction agent. Prior to any measurement, the surfaces were also cleaned with ethylene alcohol to remove any residual dirt.

The parameters of the wear test were as follows:

Applied force—0.2 N;
Ball rotational speed—80 rpm;
Speed—0.1 m/s;

Wear test time—5400 s;

Complete sliding distance—574 m.

The crater diameter was measured in two directions (parallel and perpendicular to the ball movement) with an optical microscope with an accuracy of 0.001 mm. Calculations of specific wear rates were based on the average value of crater diameter. Only craters with differences between perpendicular and parallel diameters less than 10 μm were chosen for further calculations, in compliance with PN-EN 1071-6:2007.

The wear volume was measured according to the following formula:

$$V = \frac{\beta b^4}{64R}, \tag{1}$$

where $R$ is the ball's diameter, and $b$ is the crater diameter.

The Archard equation connects wear volume, $V$, with applied force and sliding distance as follows:

$$V = k_c \cdot S \cdot N, \tag{2}$$

where $k_c$ is the specific wear rate, $S$ is the sliding distance, and $N$ is the applied force.

Electrochemical impedance spectroscopy (EIS) is often used for the determination of electrochemical processes that occur on surfaces of construction materials. It is especially useful for the determination of corrosive processes on metals, and the failure of protective coatings [20,21]. EIS is used to determine barrier properties of coating by finding the value of coating capacitance of a polymer layer of given thickness [22]. Moreover, EIS is also used to determine the influence of adducts, such as pigments and fillers, on coatings' protective properties [23]. Investigating the changes in capacitance of a 1 mm thick resin layer, cured for various exposure times, the barrier properties associated with the presence of oxidizing agents can be assessed.

To investigate the electrochemical impedance of the cured resin, each sample was cut into 1 mm thick discs and placed on a copper plate in the electrochemical unit filled with 1 mole of nitric acid. Figure 2 shows the schematic view of the unit, that consisted of the vessel, working electrode (sample), counter electrode (Pt) and reference electrode (Ag/AgCl) with Lugins capillary.

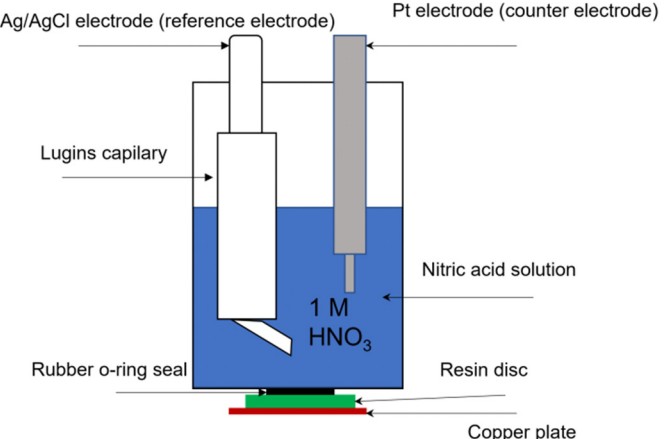

**Figure 2.** Schematic view of the electrochemical unit used for EIS tests.

Prior to the measurements, the samples were stabilized for 6 h. The impedance testing (repeated three times for each curing time) was done with a potential amplitude of 5 mV, in the frequency range from 100 kHz up to 1 MHz.

Spectra in the infrared range were taken using Spectrum Two FTIR spectrophotometer from Perkin Elmer (Waltham, MA, USA), working in the Attenuated Total Reflectance (ATR) mode. For each material, three consecutive measurements were done and averaged to obtain the final spectra.

## 3. Results and Discussion

The conducted investigations proved that initial polymerization significantly affected the hardness of the photocured polymer samples. In samples cured for 5 s, the hardness was 61.0 ± 2.4 HB. This value was 23% lower than for samples cured for 10 s and 26% lower than for samples photocured for 15 s. The difference in hardness between the last two samples was 3 HB and appeared not to be statistically significant, as presented in Table 1. Therefore, it could be assumed that it fell within measurement uncertainty.

**Table 1.** Statistical evaluation of *p*-value obtained from multiple comparisons of hardness data (Kruskall–Wallis test, $p < 0.05$ considered significant).

|  | 5 s | 10 s | 15 s |
|---|---|---|---|
| 5 s | - | 0.037 | 0.001 |
| 10 s | 0.037 | - | 0.915 |
| 15 s | 0.001 | 0.915 | - |

The difference in hardness due to curing is caused by polymerization activated by ultraviolet light. The degree of material polymerization is significant for the development of materials' physical and mechanical properties [24]. It can be assumed that preliminary curing time significantly increases values of mechanical properties, such as hardness, strength and elastic modulus. Preliminary amounts of photonic energy lead to the creation of polymerization centers, in the form of macromolecular free radicals, where elongation of polymer chains occurs. Shorter exposition to UV light results in lower polymerization efficiency which leads to less durable and softer materials [25]. Abate et al. [26] conducted research on photocurable resins, and proved the correlation between the amount of photon energy absorbed by the polymer and cured material hardness. Moreover, these authors found a constant value of standard deviation and periodical decrease and increase of hardness value. This effect suggests perpetual changes in conformation and arrangement of polymer chains during polymerization. Variations in the length and spatial orientation of the chains affect homogeneity of mechanical parameters during polymer curing until complete crosslinking occurs. Pszczolkowski and Dzadz [27] proved that hardness of the resin further increased for up to 48 days, but then it achieved a plateau. Additionally, a decrease in standard deviation values was observed. In the presented research hardness value reached a plateau of 84.8 ± 0.8 HB after 48 h of additional irradiation. This was connected with a low degree of crosslinking and lower polydispersity, which could result in lower wear resistance and hardness. The results of specific wear rates are presented in Figure 3.

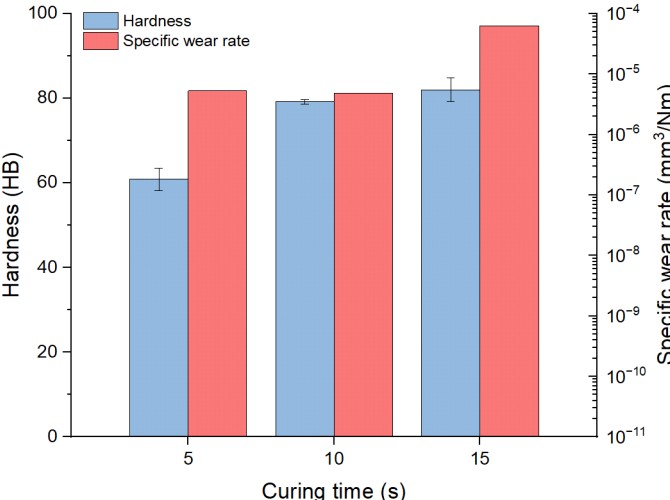

**Figure 3.** Effect of time of ultraviolet/visible light (UV/VIS) exposure on the hardness (linear scale) and specific wear rate (semi-log plot) of the resins under investigation.

The results indicated that increase in preliminary curing times significantly influenced specific wear rates. The increase in exposure times from 5 to 10 s caused a minor decrease of specific wear rate. However, an exposure time of 15 s showed the highest wear intensity values, which were about 10-fold higher than those for other samples [27]. A similar trend was previously demonstrated for polymers [28], because lower hardness corresponds to higher elastic limit, so the friction forces between elements in contact contribute to higher deformation of their materials, rather than abrasive wear.

Figure 4 shows the views of the craters created during wear investigation.

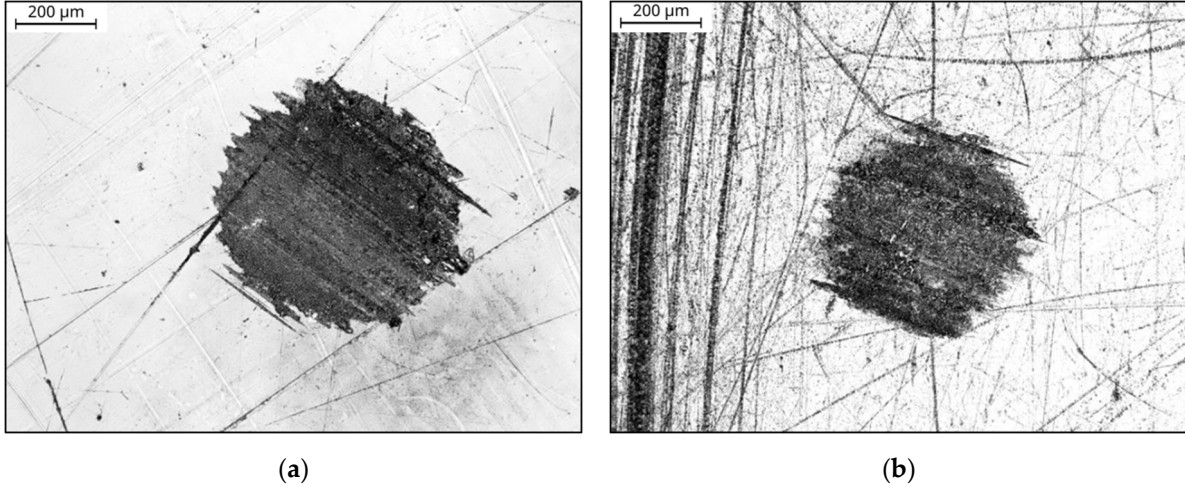

(**a**)　　　　　　　　　　　　　　　　　　(**b**)

**Figure 4.** Views of the craters created during wear investigation in polymers exposed for: (**a**) 5 s; (**b**) 15 s.

The view of a crater in the polymer sample cured for 5 s revealed deep lines, characteristic of friction wear between two materials that largely differ in hardness. Similar, although shallow, lines were seen in samples cured for 15 s. Additionally, the bottom of the crater exhibited cracks perpendicular to the movement of the counter sample (Figure 5).

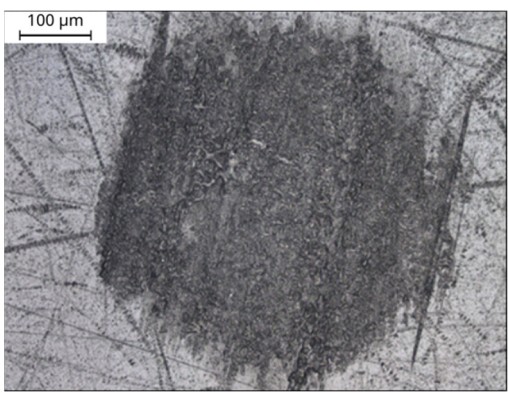

**Figure 5.** Close view of the crater in a polymer exposed for 15 s.

Higher hardness values achieved during longer exposure times caused lower elasticity of the polymer. The lowering of the elasticity threshold and lower strength of harder polymers caused movement of the surface part of the material during friction.

Research showed that preliminary curing influenced the degree of crosslinking, which correlated with the ability of the material to accumulate electric charges in the electrochemical double layer. EIS data are presented in the form of Nyquist plots, shown in Figure 6a. The obtained data could be fitted to an equivalent circuit, presented in the inset in this figure. Best-fit capacitance of the electrochemical double layer, shown in Figure 6b, revealed that irradiation time influenced the storage of electrical energy, and that this parameter turned out to be correlated with a specific wear rate.

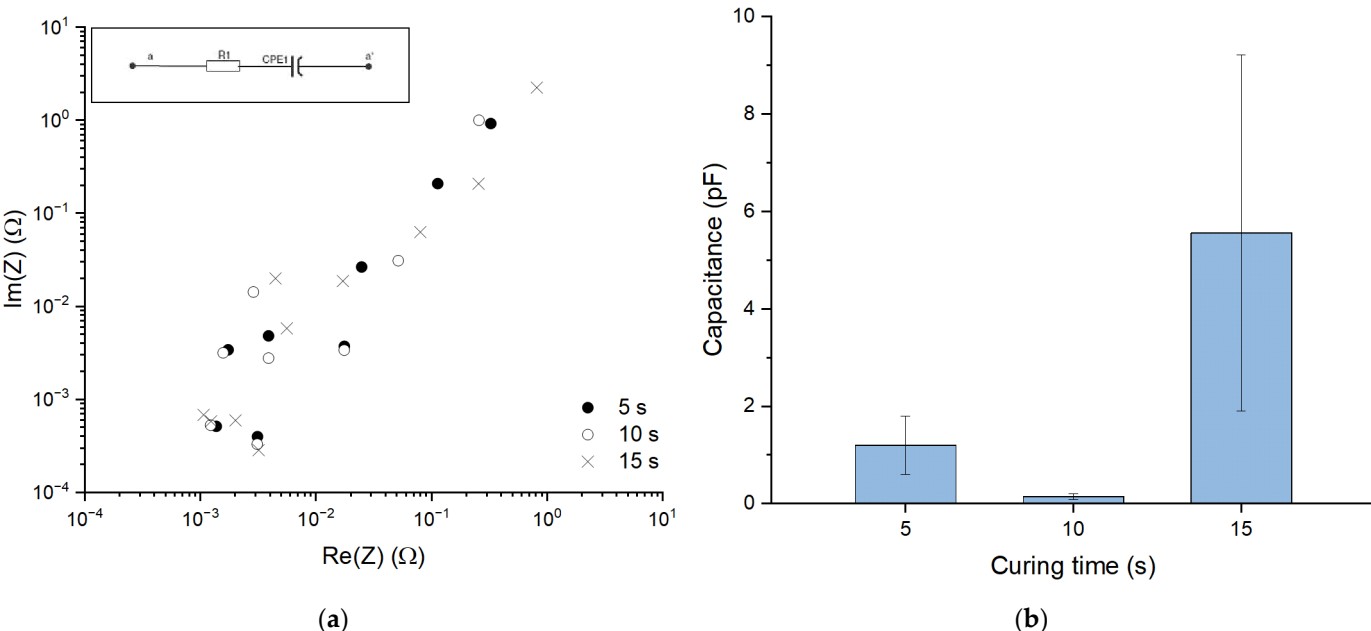

**(a)**                                                 **(b)**

**Figure 6.** Results of EIS measurements of 1 mm thick polymer layers: (**a**) log–log Nyquist plot of complex impedance (Inset: equivalent circuit used for fitting the results of EIS measurements), (**b**) best-fit serial capacitance of equivalent circuit.

Resistor $R_1$ in the equivalent circuit (inset in Figure 6a) represents the resistance of the acidic electrolyte and, therefore, its value is not a material parameter. The capacitance of the constant phase element represents the electrochemical double layer between the solution and the resin. The highest average electrochemical double layer capacitance was measured for samples cured for 15 s. The capacitance value was $5.6 \pm 3.0$ pF. In contrast, the lowest capacitance was obtained for samples cured for 10 seconds ($0.150 \pm 0.005$ pF). The decrease in capacitance under exposure to UV/VIS light for 10 s curing could be explained by the disappearance of double bonds. due to polymerization and increase in the mass of the polymer chain resulting from that process, as the presence of double conjugated bonds increases conductivity of organic material and, therefore, its capacitance [29]. The rapid increase of capacitance with elapsed exposure time could be explained by increasing selectivity of the curing that gradually affects only singular chains. This process might lead to the creation of cyclic structures, resulting in the decrease in molecular distance between carbonyl groups. Shorter distance between these groups is supposed to be proportional to the decrease of electron transfer energy, as was reported for aromatic compounds and their use in lithium batteries [30]. At the same time, the measurements of the density of the material showed little change, with values of 1.32–1.40, 1.40–1.45, 1.49–1.52 g/cm$^3$ for samples cured for, respectively, 5, 10 and 15 s.

Polymerization of acrylic monomers in composite resins leads to the creation of a highly crosslinked structure. However, monomer conversion is not usually completed and the obtained macromolecules always consist of some grafted chains [31]. These side chains may influence material density through reaction with propagating radicals in the preliminary hours of the post-production cycle. After 10 s of preliminary curing, stabilization and small standard deviation of results were observed. This irradiation period probably favored elongation of polymeric chains with a small amount of branching. These types of chain growth are called primary cycle (elongation of chain) and secondary cycle (branching of chain), respectively [32,33]. The elongation of the chain caused a decrease of material capacitance because of the increase of longitudinal dimension and decrease of molecule mobility in electric field. The branching of chains caused the increase of material capacitance because of the ionization energy decrease in tertiary carbon atoms and higher molecule mobility, in comparison to linear molecules with the same quantity of monomeric

units. The increase of capacitance for samples cured for 15 s suggested that the branching mechanism had a more prevalent role in chain growth. The increase in standard deviation confirmed observations made by Anseth and Bowman, [32], on the possibility of cyclization processes. It confirmed the change for the samples in the initial period of lighting; however, in the final period it did not confirm the theory for the 10 s case. Reference data in this aspect is scarce. However, it was suspected that during 5 s of pre-crosslinking, little nuclei of crystallization developed and the chain would grow to such an extent as to form a linear structure. The chains did not elongate uniformly due to the short curing time. The preliminary curing time of 10 s was optimal for the creation of a linear polymer chain structure, although the 15 s preliminary curing time was characterized by development of cross chains. Crosslinking, that possibly occurred at 15 s of preliminary curing, was also not a uniform process in the entire volume of the sample.

Figure 7 shows FTIR spectra of the samples under investigation.

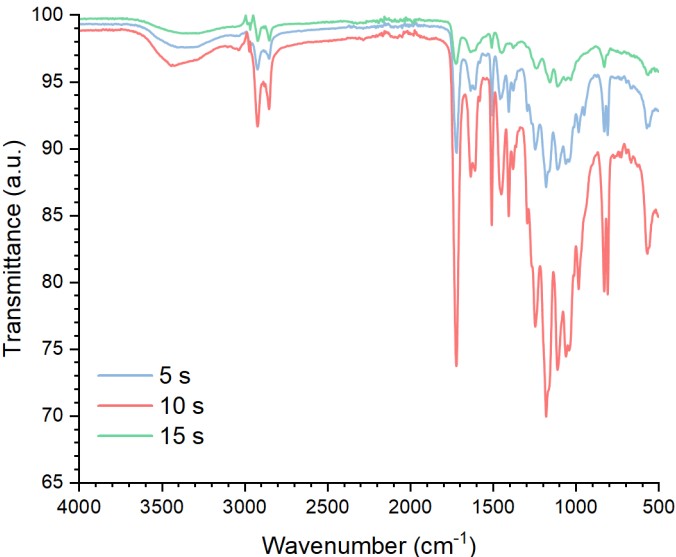

**Figure 7.** FTIR spectra taken in the ATR mode of the polymer samples under investigation.

FTIR spectra measurements were carried out to establish changes in the molecular structure of epoxy resin with elapsed curing time. The observed peaks were characteristic of acrylic epoxy resins. Regardless of curing time, neither shifts in peak positions, nor occurrence of new peaks, were observed throughout the experiments. Therefore, no chemical group appeared due to irradiation, although curing was found to modify the intensities of specific peaks. The lowest intensity of the peaks was obtained for the resin cured for 15 s, while the highest was for that cured for 5 s. The band at 3429.24 cm$^{-1}$ corresponded to O–H stretching vibrations of intermolecularly bonded hydroxy groups. The peak at 2924.63 cm$^{-1}$ indicated the presence of aldehyde groups. The bands located at 1720.50 cm$^{-1}$ represented CO stretching in acrylic acid. The peaks at 1637.02 cm$^{-1}$ indicated the presence of alkene bonds. The peak at 1509.04 cm$^{-1}$ corresponded to some kind of nitro compound. The band at 1406.52 cm$^{-1}$ indicated hydroxy groups in carboxylic acids. The peaks at 1246.61, 1181.02 and 1111.72 cm$^{-1}$ corresponded to stretching C–O–C of ethers and epoxy groups, respectively. After 5 s of curing the intensity of all peaks was significantly low and the loss of transmittance reached about 5%. Transmittance values increased after 10 s of curing, which was caused by the increase of carboxyl and hydroxy groups densities on the surface of the material. The peaks corresponding with the presence of vinyl and carboxylic groups predominantly decreased their intensity during the process of curing between 10 s and 10 s. This led to free carboxylic and hydroxy groups and an observed decrease of capacitance during the EIS tests. This result may support the theory that, during the first seconds of the printing process, elongation of chains mainly occurs due to reaction of double bonds. Between 10 s and 15 s the loss of intensity for peaks

responsible for epoxy resins occurs, which leads to a possibility of a branching mechanism being a separate reaction. Branching, based on the reaction of epoxy bonds, may lead to the increase in distances between chains.

## 4. Conclusions

The presented study demonstrated that 10 s preliminary curing with UV light leads to the creation of polymer structures with most favorable mechanical properties. Comparison of hardness, wear and EIS data led to a conclusion that circular and branched structures were created between 10 and 15 s of the preliminary curing. The results of hardness, EIS and wear rate showed that there was a sharp change in the trend for the sample cured for 15 s. On that basis, the modification in propagation mechanism through polymer chain branching may be hypothesized. Analysis of FTIR spectra concluded that up to 10 s of curing polymerization of double carbon bonds was the most prevalent mechanism, and led to more linear chain propagation. Between 10 and 15 s curing time a higher amount of epoxy groups underwent polyaddition reactions, which led to chain branching, causing a lower increase in hardness, stabilization of wear rate, and electrical capacitance, due to the creation of the branched structures. Further research could provide a deeper understanding of the interconnections between parameters of SLA 3D printing and properties of the obtained models and could lead to production optimization for selected applications.

**Author Contributions:** Conceptualization, B.P.; methodology, S.K.; software, M.L.; validation, M.B; formal analysis, S.K.; investigation, W.R., M.L.; data curation, K.L.; writing—original draft preparation, B.P.; writing—review and editing, S.K.; visualization, M.Z.; supervision, S.K., M.B. All authors have read and agreed to the published version of the manuscript.

**Funding:** This research received no external funding.

**Data Availability Statement:** Not applicable.

**Conflicts of Interest:** The authors declare no conflict of interest.

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
