# Peer review of "Beneficial Effect of Pre-Hardening of Elements Manufactured by the SLA Technology"

_lubricants, doi:10.3390/lubricants10100268_

Round 1
Reviewer 1 Report
This work selected three different curing times, i.e. 5 s, 10 s and 15 s, and mainly studied the effect of preliminary curing time on the mechanical, physicochemical and tribological properties of SLA products. The designed experiments are very easy and the results demonstrated in this work are very simple, but it lacks deep mechanism investigation. The writing and the construction of this manuscript should be improved. Since this manuscript was submitted to a special issue, it’s up to the editor to make the decision. My opinions are listed below:
(1) When the abbreviation “SLA” appears for the first time, its full name “Stereo Lithography Appearance” should be given in the Abstract/Title.
(2) In the first “Materials and Methods” part, the authors claimed that “photocurable resins exposed to 405 nm ultraviolet light”. However, in many published works, the 405 nm is considered as a visible light source. Please check some references from Prof. Jacques Lalevée, such as European Polymer Journal 2020, 132, 109737 and Polymer 2018, 159, 47–58. And what’s the intensity of the light?
(3) What’s the exact formulation of the “photocurable resins” used in this work? This should be clearly presented.
(4) An unclear scale was shown in the top left corner of Figure 3. But it’s missing in Figure 2.
(5) In this work, the authors selected three curing times: 5 s, 10 s and 15 s. But have the authors tried to use any longer curing times? Such as 20 s or 30 s?
Author Response
(1) When the abbreviation “SLA” appears for the first time, its full name “Stereo Lithography Appearance” should be given in the Abstract/Title.
A: Thank you for your comment, appropriate change has been made.
(2) In the first “Materials and Methods” part, the authors claimed that “photocurable resins exposed to 405 nm ultraviolet light”. However, in many published works, the 405 nm is considered as a visible light source. Please check some references from Prof. Jacques Lalevée, such as European Polymer Journal 2020, 132, 109737 and Polymer 2018, 159, 47–58. And what’s the intensity of the light?
A: Thank you for your comment. We agree that is information was missing in the original text. According to the manufacturer, the cross-linking of the resin occurs upon exposure to the light in the range from 365 to 405 nm.
(3) What’s the exact formulation of the “photocurable resins” used in this work? This should be clearly presented.
The brand name of the resin used throughout the studies was “Anycubic® Standard”, which was mentioned in the text.
(4) An unclear scale was shown in the top left corner of Figure 3. But it’s missing in Figure 2.
Thank you for this remark. Both figures have been corrected by adding clear scales.
(5) In this work, the authors selected three curing times: 5 s, 10 s and 15 s. But have the authors tried to use any longer curing times? Such as 20 s or 30 s?
Curing time was set up in the range from 5 to 15 s according to the manufacturer’s suggestions, but we appreciate the reviewer’s remark on longer exposure. However, it will inevitably lead to an increase in the manufacturing process beyond reasonable limit, hence we are going to check this effect in future works on the properties of cured materials.
Reviewer 2 Report
The authors tried to study some properties of a kind of 3D printed material in this work. However, there are several drawbacks and even contradictions, so I don’t think this manuscript was carefully prepared.
11. There are two Figure 2 in the manuscript.
22. The values for the wear rate in the first Figure 2 have too many 0s. It is better to use scientific notation.
33. There are no scale bars in the second Figure 2, and the scale bar cannot be clearly seen in Figure 3. Additionally, the information the authors would like to provide by the second Figure 2 and Figure 3 is not clear. I’m guessing that the authors may want to compare the crater difference with different exposure times. However, it is not clear to see the difference among them, especially that even the sizes cannot be compared because there are no scale bars. Additionally, there is no control sample to compare, such as sample without wear, in this manuscript.
44. In the materials and methods section, the authors discussed how they measured impedance, however, there is no characterizations related to impedance in the manuscript. Instead, capacitance was discussed. Apart from this error, the unit for capacitance in Figure 4 is also wrong (should be Farad).
55. It is confusing that why the authors would like to measure the capacitance of the printed polymer. Firstly, capacitance normally is a parameter of a capacitor, and it is usually size-, dimension-, and material-dependent. Secondly, the authors mentioned the 3d printed material is a kind of resin, which is not electrically conductive. Thirdly, the authors mentioned that the sample was placed on a copper plate, so I doubt that the capacitance measured here was actually the capacitance between the copper plate and the platinum counter electrode.
66. I’m guessing the authors may be trying to measure the dielectric constants of the sample. However, there is no characterization or calculations based on the capacitance measurements.
Author Response
1. There are two Figure 2 in the manuscript.
Thank you for making this comment. Numbering has been corrected.
2. The values for the wear rate in the first Figure 2 have too many 0s. It is better to use scientific notation.
We appreciate this remark. Fig. 2 has been changed accordingly.
3. There are no scale bars in the second Figure 2, and the scale bar cannot be clearly seen in Figure 3. Additionally, the information the authors would like to provide by the second Figure 2 and Figure 3 is not clear. I’m guessing that the authors may want to compare the crater difference with different exposure times. However, it is not clear to see the difference among them, especially that even the sizes cannot be compared because there are no scale bars. Additionally, there is no control sample to compare, such as sample without wear, in this manuscript.
Numbering of figures has been updated, so the scale bars in them. As to control sample, there is enough uncovered area in both Fig. 4a and Fig 4b to see the surface of a sample not affected by the wear test. In our opinion no additional reference sample is necessary for this purpose.
4. In the materials and methods section, the authors discussed how they measured impedance, however, there is no characterizations related to impedance in the manuscript. Instead, capacitance was discussed. Apart from this error, the unit for capacitance in Figure 4 is also wrong (should be Farad).
We greatly appreciate this remark. Indeed, we have corrected wrong units and introduced Nyquist plot and showed equivalent circuit. Furthermore, short discussion of obtained results have been added to the text.
5. It is confusing that why the authors would like to measure the capacitance of the printed polymer. Firstly, capacitance normally is a parameter of a capacitor, and it is usually size-, dimension-, and material-dependent. Secondly, the authors mentioned the 3d printed material is a kind of resin, which is not electrically conductive. Thirdly, the authors mentioned that the sample was placed on a copper plate, so I doubt that the capacitance measured here was actually the capacitance between the copper plate and the platinum counter electrode.
As we mention in the original text, EIC does not measure the capacitance of the resin (depending on the area, disc thickness and its permittivity), but that of the double layer between the solution and the polymer. From that, we may conclude on the dynamics of polymerization in terms of the presence/absence of specific bonds, re-arrangement of molecular structures etc.
6. I’m guessing the authors may be trying to measure the dielectric constants of the sample. However, there is no characterization or calculations based on the capacitance measurements.
Indeed, we have calculated parameters of equivalent serial RC circuit rather than measure permittivity of the polymer. However, in the updated manuscript we enclosed some data on the capacitance.
Reviewer 3 Report
1. Many publications exist where the researchers study the curving parameters and their influence on the mechanical properties. The authors should rewrite the introduction and pay attention to these studies. Now the introduction is not related to the topic of the article.
2. Provide the resin manufacture, ball materials, and model of tribometer.
Table 1 does not contain the unit and name of the parameters.
3. What does this table show? Why some points are missed?
4. Why did the authors choose 15 sec? What happens with the polymerization during the 20, 30, 60 sec?
5. Why the impedance was studied? From my point of view, it is not important information. Please provide the impedance curves. It is very useful to provide the curve for a long time of polymerization (1 min or more).
6. Better start the discussion from the FTIR spectra and after the mechanical and electrical properties. Provide the FTIR for the time >1 min. FTIR spectra should be normalized by maximum for the correct comparison. Raman spectra also can provide additional information.
Author Response
1. Many publications exist where the researchers study the curing parameters and their influence on the mechanical properties. The authors should rewrite the introduction and pay attention to these studies. Now the introduction is not related to the topic of the article.
Our work aimed at assessing some mechanical properties and uncover the dynamics of cross-linking within polymers made of easily accessible (although trademarked, hence not fully uncovered) resin using popular additive manufacturing technique (SLA). To place our work in the wider context we gave several references to similar works in this field. We agree that there is a bunch of other works, but to our good faith none of them covers the topic and the methods used in our paper.
2. Provide the resin manufacturer, ball materials, and model of tribometer.
This information was updated in the revised manuscript.
3. Table 1 does not contain the unit and name of the parameters. What does this table show? Why some points are missed?
Table 1 shows the unitless p-values obtained to prove the significance of differences among three hardness data due to various curing times. As is stated in the caption, p less than 0.05 means that compared quantities differ significantly due to the treatment method, otherwise this difference is considered insignificant. That is why diagonal cells are empty.
4. Why did the authors choose 15 sec? What happens with the polymerization during the 20, 30, 60 sec?
Curing time was set up in the range from 5 to 15 s according to the manufacturer’s suggestions, but we agree as to longer exposure. However, it will increase the manufacturing process beyond reasonable limit, hence we are going to check this effect in future works on the properties of cured materials.
5. Why the impedance was studied? From my point of view, it is not important information. Please provide the impedance curves. It is very useful to provide the curve for a long time of polymerization (1 min or more).
The impedance was studied because it offers insight into the dynamics of cross-linking in the polymer observing the serial RC circuit formed as a double layer. After all, the Nyquist plot (in the double-log plot form) is added to the revised text.
6. Better start the discussion from the FTIR spectra and after the mechanical and electrical properties. Provide the FTIR for the time >1 min. FTIR spectra should be normalized by maximum for the correct comparison. Raman spectra also can provide additional information.
Thank you for this comment, however, it was impossible to make any Raman scattering experiments with these samples. After all, FTIR spectra have been normalized accordingly.
Round 2
Reviewer 1 Report
After the appropriate revision, this manuscript may be accepted for publication.
Author Response
Dear Reviewer,
We appreciate your effort towards polishing the quality of our manuscript. Thank you for your comments.
Reviewer 2 Report
The authors made some modifications to address the comments from the reviewers. Nonetheless, there are some suggestions:
1. Figure 3:
Why the specific wear rates increase with hardness? I think the wear rate will decrease with hardness increasing.
2. Figure 6b,
The authors provided some explanations about the capacitance of the three samples. Please also provide some references about it. In addition, the authors mentioned that “The rapid increase of capacitance after further curing can be explained by closing of curing within singular chains. This process may lead to the creation of ring structures resulting in the decrease in molecular distance between car-bonyl groups.” Do the sample have a large difference in terms of the density? Why not use dielectric constant instead of capacitance?
Author Response
Q1: Why the specific wear rates increase with hardness? I think the wear rate will decrease with hardness increasing
Observed dependency was demonstrated for polymers: lower hardness corresponds to higher elastic limit, so the friction forces between elements in contact contribute to higher deformation of their materials rather than abrasive wear. See for example: 10.5604/01.3001.0010.6975. Appropriate explanation has been included in the manuscript.
Q2: The authors provided some explanations about the capacitance of the three samples. Please also provide some references about it. In addition, the authors mentioned that “The rapid increase of capacitance after further curing can be explained by closing of curing within singular chains. This process may lead to the creation of ring structures resulting in the decrease in molecular distance between car-bonyl groups.” Do the sample have a large difference in terms of the density? Why not use dielectric constant instead of capacitance?
The text has been changed accordingly and two references have been added:
"The decrease in capacitance under exposure to UV/VIS light for 10 s curing can be explained by the disappearance of double bonds due to polymerization and increase in the mass of the polymer chain resulting from that process, as presence of double conjugated bonds increases organic material conductivity and therefore capacitance [Ref1]. The rapid increase of capacitance after further curing can be explained by closing of curing within singular chains. This process may lead to the creation of cyclic structures resulting in the decrease in molecular distance between carbonyl groups. Shorter distance between those groups is supposed to be proportional to the decrease of electron transfer energy as was reported for aromatic compounds and their use in lithium baterries [Ref2]
Ref1. Thanh-Hai Le, Yukyung Kim, Hyeonseok Yoon 1,2,Electrical and Electrochemical Properties of Conducting Polymers, polymers, 2017, 9, 150, doi:10.3390/polym9040150
Ref2. Huiling Peng, Qianchuan Yu, Shengping Wang, Jeonghun Kim, Alan E. Rowan, Ashok Kumar Nanjundan, Yusuke Yamauchi, and Jingxian Yu, Molecular Design Strategies for Electrochemical Behavior of Aromatic Carbonyl Compounds in Organic and Aqueous Electrolytes, Adv.Sci., 2019, 6, 1900431"
Apart from that, the measurements of density have shown little change:
1.32 – 1.40 g*cm-3 for samples cured for 5s
1.40 – 1.45 g*cm-3 for samples cured for 10s
1.49 – 1.52 g*cm-3 for samples cured for 15s
The parameters were chosen to determine electrochemical properties of the material rather than electrical. Research were partially conducted to assess durability of the material in aggressive chemical environment, hence the use of HNO3 as the solution.
Reviewer 3 Report
My suggestion was taken into account.
Author Response

(The authors gave the same response as above.)
